



# Parameterizing the aerodynamic effect of trees in street canyons for the street-network model MUNICH using the CFD model Code_Saturne

Alice Maison[1,2], Cédric Flageul[3], Bertrand Carissimo[1], Andrée Tuzet[2], Yunyi Wang[1], and Karine Sartelet[1]

[1]CEREA, École des Ponts, EDF R&D, Marne-la-Vallée, France
[2]Université Paris-Saclay, INRAE, AgroParisTech, UMR EcoSys, 78850 Thiverval-Grignon, France
[3]PPRIME Institute, Curiosity Group, Université de Poitiers, CNRS, ISAE-ENSMA, France

**Correspondence:** Alice Maison (alice.maison@enpc.fr), Karine Sartelet (karine.sartelet@enpc.fr)

**Abstract.** Trees provide many ecosystem services in cities such as urban heat island reduction, water runoff limitation and carbon storage. However, the presence of trees in street canyons reduces the wind velocity in the street and limits pollutant dispersion. Thus, to get accurate simulations of pollutant concentrations, the aerodynamic effect of trees should be taken into account in air-quality models at the street level.

The Model of Urban Network of Intersecting Canyons and Highways (MUNICH) simulates the pollutant concentrations in a street-network, considering dispersion and physico-chemical processes. It can be coupled to a regional-scale chemical-transport model to simulate air quality over districts or cities. The aerodynamic effect of tree crown is parameterized here, through their impacts on the average wind velocity in the street direction and the vertical transfer coefficient associated with the dispersion of a tracer. The parameterization is built using local-scale simulations performed with the Computational Fluid Dynamics

(CFD) code Code_Saturne. Two-dimensional CFD simulations in an infinite street canyon are used to quantify the effect of trees depending on the tree characteristics (Leaf Area Index, crown volume fraction and tree height to street height ratio) using a drag-porosity approach. The tree crown slows down the flow and produces turbulent kinetic energy in the street, thus impacting the tracer dispersion. This effect increases with the Leaf Area Index and the crown volume fraction of the trees: the average horizontal velocity in the street is reduced up to 68 % and the vertical transfer coefficient up to 23 % in the simulations

performed here.

A parameterization of these effects on horizontal and vertical transfers for the street model MUNICH is proposed. Existing parameterizations in MUNICH are modified based on Code_Saturne simulations to account for both building and tree effects on vertical and horizontal transfers. The parameterization is built to get similar tree effects (quantified by a relative deviation between the cases without and with trees) between Code_Saturne and MUNICH. The vertical wind profile and mixing length

depend on Leaf Area Index, crown radius and tree height to street height ratio. The interaction between the trees and the street aspect ratio is also considered.



## 1 Introduction

Cities are, by definition, areas with high density of people, infrastructures and activities, and this urbanization causes many issues. First, air quality is poor because of the numerous air pollutants emitted by anthropic activities such as traffic, indus-
tries or residential activities, and the reduction of air flow by high buildings limits the dispersion of these pollutants (Faiz, 1993; Akimoto, 2003; Yuan et al., 2014; Zhang et al., 2020). In addition to air pollution issues, radiative and water budgets are strongly modified in cities compared to countryside (Bozonnet et al., 2015). Temperatures are, in average, higher than in the countryside because of the urban heat island created by additional anthropogenic energy released, storage of radiative energy in dark materials, radiation multireflection and lack of vegetation and associated evapotranspiration (Oke, 1982; Pigeon
et al., 2007; Stewart, 2011; Hebbert and Jankovic, 2013). Impervious soils also decrease water infiltration and intensify runoff (Leopold, 1968). In addition, growing urbanisation and increasing extreme events (due to climate change) such as pollution peaks, heat waves and floods have negative consequences on environment and human health (Robine et al., 2007; Angel et al., 2011; Pascal et al., 2013; West et al., 2016; IPCC, 2021).

One nature-based solution is to green the city by planting vegetation as lawns, trees in streets or in parks, green walls and roofs
(Livesley et al., 2016; Revelli and Porporato, 2018). Vegetation and especially trees contribute to improve human thermal comfort by creating favorable micro-climate with lower air temperature (through solar radiation interception and creation of shade) and higher evaporation (through ground and vegetation evapotranspiration) (Taha et al., 1991; Bowler et al., 2010; Gillner et al., 2015; Klemm et al., 2015; Lobaccaro and Acero, 2015; Gunawardena et al., 2017). This positive effect of trees is significant in particular during heat wave episodes, which will be more frequent in the future due to climate change (IPCC, 2021). Trees and
vegetated areas also favorize infiltration in soils that contributes to offset water runoff induced by soil artificialization (Armson et al., 2013; Berland et al., 2017). Besides, vegetation is known to store carbon (Nowak and Crane, 2002; Svirejeva-Hopkins et al., 2004) and to enhance human well-being (van Dillen et al., 2012; Bertram and Rehdanz, 2015; Krekel et al., 2015). For all these ecosystem services in urban areas, city greening is often promoted and for example the city of Paris has about 205,000 trees of which 52 % are roadside trees (Municipality of Paris, 2021).

Many studies have tried to figure out the impact of trees on air pollution in a street canyon and they have shown that trees are an important parameter to take into account if we want to understand and simulate accurately the pollutant concentrations in the streets (Beckett et al., 1998; Nowak et al., 1998, 2006; Jayasooriya et al., 2017). Vegetation and especially trees represent surfaces available for pollutant dry deposition, and hence can contribute to reduce air pollutants concentrations. However, pollutant removal and its impact on air quality vary greatly depending on tree characteristics, tree species and pollutant type
(Hwang et al., 2011; Selmi et al., 2016; Xue and Li, 2017; Ozdemir, 2019). Trees may alter air quality by influencing aerodynamic processes and limiting the pollutant dispersion (Buccolieri et al., 2011; Wania et al., 2012; Vos et al., 2013; Gromke and Ruck, 2007, 2009, 2012; Gromke and Blocken, 2015). The aerodynamic effect of trees is defined as the drag force resulting from the friction between the air and the leaves. Since the tree crown can be seen as a porous medium, as air passes through it but is slowed down and the drag force increases with the leaf surface.

For a wind perpendicular to a street, the air recirculate inside the street canyon (Harman et al., 2004). Pollutants emitted at the





bottom of the street (by traffic) accumulate on the leeward side of the street inducing higher local concentrations (Vardoulakis et al., 2003; Cai et al., 2008; Huang et al., 2019). Obstacles in the street, such as trees, can strongly impact the air flow and intensify the pollutant accumulation (Vos et al., 2013). This effect has been studied using Computational Fluid Dynamics (CFD) models where the aerodynamic influence of trees on the flow is represented by a porosity model (Buccolieri et al., 2009; Zaïdi

et al., 2013; Wei et al., 2016; Jeanjean et al., 2017; Santiago et al., 2017). Vos et al. (2013) showed that pollutant concentrations may increase by 20 % in a street, because of the presence of two rows of trees in the street. The effects of vegetation depend on the height, width and density (Leaf Area Index) of trees (Vos et al., 2013; Janhäll, 2015), as well as the height to width ratio of streets (Wania et al., 2012). Finally, it is necessary to assess accurately the effect of trees on pollutant dispersion in order to find what configurations are more effective in reducing air pollution in street canyon and to guide urban development policy

(Janhäll, 2015).

The effect of trees on aerodynamic processes should also be considered at the street level in air quality models. As discussed previously, CFD models including trees are used to study wind fields and pollutant transport in street canyons (Li et al., 2006). However, as the street is discretized with a fine mesh, the computational cost is high and simulations at the city scale are too expensive today. Fast-running codes such as simplified street-network or street-in-grid models are developed to simulate street

pollutant concentrations over neighbourhood or cities, but they do not take into account the effect of trees in the streets. The objective of this study is to parameterize the effect of trees on air flow in the Model of Urban Network of Intersecting Canyons and Highways (MUNICH) (Kim et al. (2018); Lugon et al. (2019); http://cerea.enpc.fr/munich/, last access on 17 December 2021). To build this parameterization, simulations in street canyons are performed with Code_Saturne (Archambeau et al. (2004); https://www.code-saturne.org/, last access on 17 December 2021), a CFD code, which can represent the tree aerody-

namic effect with a drag porosity approach (Katul et al., 2004) and has previously been compared with field measurements (Zaïdi et al., 2013). MUNICH parameterizations have already been compared with Code_Saturne results in a treeless canyon and a new parameterization for horizontal and vertical transfers have been developed in MUNICH based on Wang (2012, 2014) and Code_Saturne simulations (Maison et al., 2022). In the present study, the tree aerodynamic effect is added to this parameterization and Code_Saturne (version 6.0) is used as a reference to parameterize the aerodynamic effect of trees in the street

network model MUNICH. CFD simulations are performed in three streets of aspect ratios varying from 0.3 to 1.0 and a large range of tree leaf area index, crown radius and heights is tested. The tree aerodynamic effect quantified with Code_Saturne is analysed depending on these street and tree characteristics.

The approach to model the dispersion of pollutants in MUNICH and Code_Saturne is fundamentally different due to their physical modeling and discretization in space and time. MUNICH is a street-network model that simulates air pollutant con-

centration in a urban canopy. Street dimensions and pollutant concentrations are assumed to be homogeneous in each street segment. Air flow is divided into horizontal flux from one street to another, and vertical flux between the street and the background (Maison et al., 2022). Background concentrations above the street can be computed by 3D Chemistry-transport models (CTMs), such as Polair3D (Sartelet et al., 2018; Lugon et al., 2019, 2021). To build the tree parameterization, the CFD simulation setup is adapted for the comparison with MUNICH and several simulations are performed with Code_Saturne considering

a range of street and tree characteristics.





The structure of the paper is as follows. The MUNICH and Code_Saturne models are presented in Sect. 2. Then, the tree effect on horizontal wind speed and vertical transfer coefficient is quantified with Code_Saturne simulations in Sect. 3 and parameterized in MUNICH in Sect. 4. Conclusions are presented in Sect. 5.

## 2   Materials and methods

### 2.1   Description of MUNICH

In MUNICH, each street segment is assumed to be homogeneous, i.e. with uniform building height $H$ and street width $W$, and of length $L$ (m). The street is characterized by its height-to-width ratio called aspect ratio, $a_r = H/W$ (-). Only the average pollutant concentrations over the street are considered. Pollutants are transported by the horizontal wind speed (advection) in the street network and by a vertical transfer coefficient between the streets and the background. Several parameterizations of

the horizontal wind speed and of the vertical transfer coefficient exist in MUNICH. The ones recently developed in Maison et al. (2022) and based on Code_Saturne simulations are used and detailed here. The vertical profile of the wind speed in the street direction is calculated as an attenuation of the wind speed in the street direction and at the roof level, $U_{H,\varphi}$ (m.s$^{-1}$) as (Maison et al., 2022):

$$U(z) = U_{H,\varphi} \left[ C_1 I_0(g(z)) + C_2 K_0(g(z)) \right] \quad \text{with } U_{H,\varphi} = U_H |\cos(\varphi)|, \tag{1}$$

where $C_1$ and $C_2$ are integration coefficients, $I_0$ and $K_0$ are the first and second type modified Bessel functions of order 0. Besides, the wind speed at the roof level $U_H$ has to be multiplied by $|\cos(\varphi)|$ to select the component of the wind speed in the street direction. This vertical wind profile is then integrated between the soil roughness $z_{0_s}$ (m) and $H$ to compute the average horizontal wind speed in the street direction. The function $g$ is calculated as (Wang, 2012, 2014):

$$g(z) = 2\sqrt{\alpha \frac{z}{H}}, \tag{2}$$

and:

$$C_1 = \frac{1}{I_0(g(H)) - I_0(g(z_{0s})) K_0(g(H))/K_0(g(z_{0s}))} \quad \text{and} \quad C_2 = -\frac{C_1 I_0(g(z_{0s}))}{K_0(g(z_{0s}))}, \tag{3}$$

where $\alpha$ is a dimensionless coefficient expressing the effects of wind angle on wind attenuation in the street, and it is computed as:

$$\alpha = \frac{C_B a_r}{\kappa s_H} \quad \text{with } C_B = 0.31 \left[ 1 - \exp(-1.6 a_r) \right] f_\varphi \tag{4}$$

$$\text{and } f_\varphi = \begin{cases} |\cos(2\varphi)|^3 & \text{if } \varphi \in [0, 45°] \cup [135, 225°] \cup [315, 360°] \\ 0 & \text{if } \varphi \in \,]45, 135°[ \,\cup\, ]225, 315°[, \end{cases} \tag{5}$$

where $s_H = s(z = H)$ is a dimensionless factor describing the effect of canopy on the mixing length $l_m$ (m) (Wang, 2012, 2014). The mixing length is calculated as:

$$\frac{1}{l_m} = \frac{1}{\kappa z} + \frac{1}{l_{c_b}} \implies l_m = \kappa z \frac{l_{c_b}}{l_{c_b} + \kappa z} = \kappa z \, s(z), \tag{6}$$





where $\kappa z$ corresponds to the mixing length over a rough bare soil (without canopy), and $l_{c_b}$ is the characteristic length (m) in
the street canyon, corresponding to the mixing length of the urban canopy alone ($l_{c_b} = 0.5\,W$).

The vertical transfer coefficient that drives pollutant exchange between the street and the background zone is calculated at
$z = H$ as:

$$q_{vert} = \sigma_W\, l_m = \sigma_W\, \kappa H\, s_H, \tag{7}$$

where $\sigma_W$ (in m.s$^{-1}$) is a velocity scale equal to the standard deviation of the vertical wind speed. It depends both on the
friction velocity above the urban canopy, $u_*$, and on atmospheric stability (Soulhac et al., 2011).

In section 4, the tree effect is parameterized by taking into account the characteristic length of the trees in the mixing length
$l_m$ (Eq. (19), which leads to modifications in the coefficient $\alpha$ Eq. (23)).

## 2.2    Description of Code_Saturne

### 2.2.1    Street and tree modeling setup

In the present study, the tree effect is studied in three street canyons of different street aspect ratios: a wide street canyon (WC),
an intermediate canyon (IC) and a narrow canyon (NC). Their characteristics are presented in Table 1:

**Table 1.** Characteristics of the three canyons studied.

| Canyon | Building height $H$ (m) | Street width $W$ (m) | Street aspect ratio $a_r$ (-) | Maximum height of the domain (m) |
|--------|--------|--------|--------|--------|
| WC | 8.5 | 27.5 | 0.3 | 25.5 |
| IC | 14.0 | 27.5 | 0.5 | 42.0 |
| NC | 27.5 | 27.5 | 1.0 | 82.5 |

A color code is introduced to simplify the reading of the figures: red symbols for the wide canyon (WC), purple ones for the
intermediate canyon (IC) and blue ones for the narrow canyon (NC). Abbreviations, parameters and variables used are listed
in Appendix A.

The $k - \varepsilon$ linear production turbulence model is used in Code_Saturne. Stationary simulations are performed with a thermally
neutral atmosphere. A 2D infinite street canyon is modeled with periodic condition in $Y$ axis but the flow and the wind speed
vector are 3D. The mesh is composed of hexahedral cells of 1 m in $Y$ axis direction and 0.5 m in $X$ and $Z$ axis. The vertical
profiles of $U$, $k$ and $\varepsilon$ are set in inlet (top left border of the domain). The complete description of Code_Saturne simulation
setup in treeless canyons can be found in Maison et al. (2022).

The tree geometry and its representation in Code_Saturne is shown in Fig. 1. Here, two rows of trees ($n = 2$) are considered,
one on each side of the street, whose positions on the $X$ axis are $X = 34.5$ m and $X = 48.0$ m (the wall positions are $X = 27.5$
and $X = 55.0$ m). $r$ is the tree radius (m), $h$, $h_{min}$ and $h_{max}$ correspond respectively to the middle, minimum and maximum
heights of the tree crown (m). Note that all the simulations verify $h_{max} \leq H$, i.e. the top of the trees do not exceed the top of
the street.





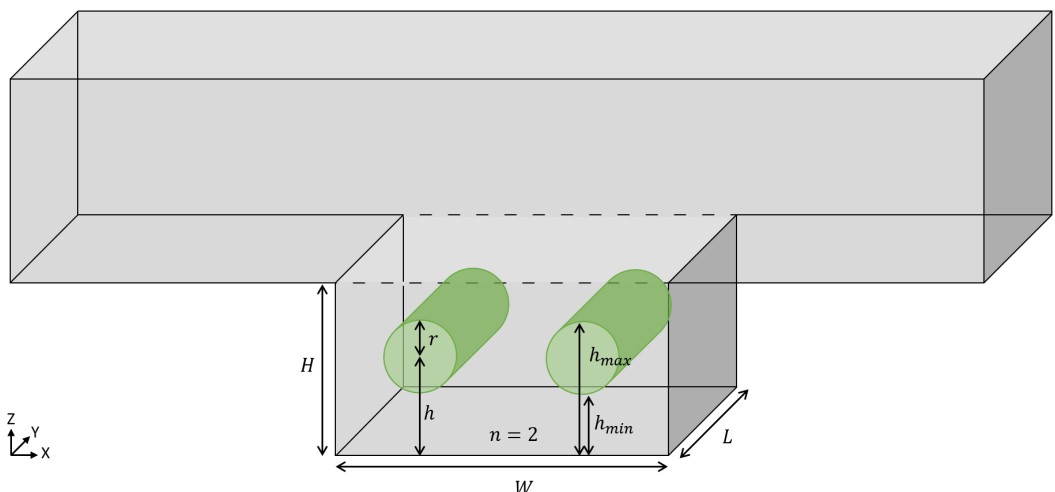

**Figure 1.** 3D scheme of the street canyon with cylindrical tree crown dimensions.

In the cells containing the trees, an additional drag term is added to the Navier-Stokes equations (Katul et al., 2004; Zaïdi et al., 2013):

$$S_{u,i} = -\rho LAD \, C_{D_t} |U| \overrightarrow{U_i}, \qquad (8)$$

where $|U|$ is the modulus of U, $U_i$ is the velocity in the i-direction, $LAD$ is the Leaf area density in $\mathrm{m^2_{leaves}.m^{-3}_{tree\ crown}}$, $\rho$ is the air density and $C_{D_t}$ is the tree drag coefficient set to 0.2, a representative value for trees (Katul and Albertson, 1998). The Leaf Area Index ($LAI$) is the one-sided green leaf area per unit ground surface area and the Leaf Area Density ($LAD$) is the one-sided green leaf area per unit volume. They are calculated as:

$$LAI = \frac{\text{surface of leaves}}{\text{soil projected surface area}} \; \left(\mathrm{m^2_{leaf}.m^{-2}_{soil}}\right) \; \text{ and } \; LAD = \frac{\text{surface of leaves}}{\text{volume of tree crown}} \; \left(\mathrm{m^2_{leaf}.m^{-3}_{crown}}\right). \qquad (9)$$

Source terms for trees are also implemented in the $k - \varepsilon$ equations:

$$S_k = \rho LAD \, C_{D_t} \left(\beta_p |U|^3 - \beta_d |U| k\right) \qquad (10)$$

$$S_\varepsilon = \rho LAD \, C_{D_t} \left(C_{4\varepsilon} \beta_p |U|^3 \frac{\varepsilon}{k} - C_{5\varepsilon} \beta_d |U| \varepsilon\right), \qquad (11)$$

where $C_{4\varepsilon} = C_{5\varepsilon} = 0.9$, $\beta_p = 1.0$ and $\beta_d = 5.03$ are constants of the model (Zaïdi et al., 2013). Note that this type of tree aerodynamic effect modeling is commonly used and evaluated by comparison with experimental results (Buccolieri et al., 2018).

Only the impact of the tree leaves is considered and the impact of the tree trunk and branches on the flow is not modeled. As the streets are modeled in 2D and an infinite length in the $Y$ direction is assumed (see section 2), the $LAI$ considered here is the equivalent cylindrical $LAI_{2D}$. Usually, studies consider the tree $LAI$, which is well defined for a 3D tree. The relation between the tree $LAI_{3D}$ and the equivalent cylindrical $LAI_{2D}$ is given in Appendix B. To check that this effect of trees is





well modeled using this simplified 2D setup, 3D simulations were performed and compared to some of the 2D simulations presented here, showing very good agreement between the 2D and 3D simulations in terms of the influence of the trees on the
flow (Appendix B).

To get a complete vision of the effect of trees, 45 simulations are performed per canyon, and the impact of three independent tree parameters are studied:

- the 2D equivalent LAI: $LAI_{2D} = 0.5, 1, 2, 3$ and $4$ (see Appendix B for the conversion between $LAI_{3D}$ and equivalent $LAI_{2D}$).

- The Crown Volume Fraction ($CVF$) calculated as the tree (2D cylindrical) crown volume divided by the street volume: $CVF = \dfrac{n\pi r^2}{HW}$. Ratio already used in studies quantifying the effect of trees on street pollution such as Gromke and Blocken (2015), (see Appendix B for the relation between 2D and 3D $CVF$). Note that apart from the Appendix B, all $CVF$ mentioned in the figures and in the text refer to the 2D $CVF$. Three $CVF$ ranges are simulated: $CVF \approx 5, 10$ and $25$ %.

- The tree to street height ratio calculated as the crown middle height divided by the building height ($h/H$). Three $h/H$ ranges are simulated: $h/H \approx 1/3, 1/2$ and $2/3$.

Note that these three tree characteristics are normalized by street characteristics, and similar values are chosen in the three canyons to be able to compare them and to quantify if the street aspect ratio influences the tree effect (i.e. if there is an interaction between tree and building effects). However, the normalized tree characteristics are not exactly equal in the three
canyons due to the street sizes and the $0.5$ m mesh cells that limit the possible range of tree sizes. The detailed list of the tree characteristics and the different values used for the tree parameters are presented in Table A4.

### 2.2.2 Calculation of vertical and horizontal transfers in Code_Saturne for comparison to MUNICH

To evaluate the vertical transfer between the street and the background zone in Code_Saturne, a passive tracer is emitted in each mesh cell of the street with an arbitrary stationary emission rate $e = 1000 \, \mu g.s^{-1}$ for a street canyon of length $L = 1$ m. For the
tracer to be dispersed only by vertical transfers and not by horizontal winds within the street, simulations are performed with wind perpendicular to the street at the inlet ($\varphi = 90$ °). The initial street and background tracer concentrations are zero. At the end of the simulation (stationary state reached), the tracer concentration is averaged in the street and in the background zone (noted $C_{street}$ and $C_{bg}$ in $\mu g.m^{-3}$) to reproduce MUNICH homogeneous street assumption. The background zone corresponds to the area of the same volume as the street but located just above the street (see Maison et al. (2022) for more details). In
absence of other processes (horizontal transport, deposition and chemical reactions), the tracer mass balance in the street yields to:





$$Q_{vert} = e \tag{12}$$

$$\implies q_{vert} \, WL \, \frac{C_{street} - C_{bg}}{H} = e \tag{13}$$

$$\implies q_{vert} = \frac{eH}{WL \, (C_{street} - C_{bg})}, \tag{14}$$

where $Q_{vert}$ is the vertical flux of pollutant at the roof level for the whole street ($\mu$g.s$^{-1}$), $q_{vert}$ is the vertical transfer coefficient (m$^2$.s$^{-1}$) and $WL$ is the exchange surface (m$^2$). Thus, the vertical transfer coefficient can be compared between both models: in Code_Saturne, $q_{vert}$ is calculated from the emission rate $e$ and the concentration gradient $\left( \frac{C_{street} - C_{bg}}{H} \right)$ following Eq. (14), and in MUNICH it is calculated from Eq. (7).

In MUNICH, the horizontal transfer velocity is equal to the street average horizontal wind speed in the street direction $U_{street}$, which is calculated by integrating Eq. (1) between $z = z_{0_s}$ and $z = H$. In Code_Saturne, $U_{street}$ is estimated from the wind speed in the $Y$ direction averaged over the street mesh cells. In addition to $U_{street}$, the MUNICH vertical profile of wind speed (Eq. (1)) can also be compared to Code_Saturne by averaging the wind speed in the $Y$ direction over the street width ($X$ axis). As the objective of the study is to parameterize the aerodynamic effect of trees, chemistry and deposition on built and vegetated surfaces are not considered here. Further details on the simulations setup of MUNICH and Code_Saturne and on the comparison of vertical and horizontal transfers in a treeless canyon are presented in Maison et al. (2022).

## 3 Quantification of the tree crown effect on horizontal and vertical transfers by Code_Saturne simulations

To quantify tree effect on the horizontal wind speed along the street and on the vertical transfer coefficient, Code_Saturne simulations are performed for a large range of tree characteristics ($LAI_{2D}$, $CVF$ and height ratio $h/H$), as summarized in Table A4. The tree effect is expressed as a relative deviation between the simulations without and with trees.

### 3.1 Tree effect on horizontal transfer

To quantify and compare the effect of tree crowns in Code_Saturne and MUNICH, and thus to overcome eventual differences between the two models observed in a treeless canyon for the horizontal velocity $U_{street}$, a relative deviation (RD) of $U_{street}$, between the simulations with and without tree, is computed as:

$$RD_{U_{street}} = \frac{U_{street} - U_{street_0}}{U_{street_0}}, \tag{15}$$

where $U_{street_0}$ stands for the average wind velocity in a treeless street and $U_{street}$ is the average wind velocity in a street with trees, as computed with Code_Saturne. This RD between the Code_Saturne simulations with and without tree is shown in Fig. 2 (a) for WC, (b) for IC and (c) for NC. It shows that $RD_{U_{street}}$ becomes increasingly negative meaning that $U_{street}$ decreases as the $LAI_{2D}$ and the $CVF$ increase. On the opposite, the tree height has either no impact or a small impact compared to $LAI_{2D}$ and $CVF$. When the effect of tree height is noticeable, an increase of the tree height induces a decrease of RD.





In the range of the tree characteristics studied, $U_{street}$ is attenuated from 7.3 to 62.3 %. The tree effect on $U_{street}$ can be compared between the three canyons. The tree effect on $U_{street}$ increases as the canyon is deeper, highlighting the complex interaction between the street dimensions and the tree effect on the velocity. This observation is consistent with the study of Wania et al. (2012).

**Figure 2.** Relative deviation RD of $U_{street}$ computed from Code_Saturne simulations for different tree $LAI_{2D}$, $CVF$ and height ratio for WC (a), IC (b) and NC (c). The graphic is divided into three columns corresponding to the three height ratios, and higher $CVF$ correspond to darker colors. $|RD_{U_{street}}|$ values are specified with data labels.





## 3.2 Tree effect on vertical transfer

The relative deviation of the vertical transfer coefficient ($RD_{q_{vert}}$) between simulations with and without tree is introduced to quantify the tree effect on the vertical transfer coefficient. Similarly to the relative deviation of $U_{street}$ (Eq. (15)), the relative deviation $RD_{q_{vert}}$ is expressed as:

$$RD_{q_{vert}} = \frac{q_{vert} - q_{vert_0}}{q_{vert_0}},\qquad(16)$$

where $q_{vert_0}$ stands for the vertical transfer coefficient in a treeless street and $q_{vert}$ is the vertical transfer coefficient in a street

with trees. Code_Saturne $RD_{q_{vert}}$ is plotted for different tree parameters $LAI_{2D}$, $CVF$ and height ratio $h/H$ in Fig. 3 (a) for WC, 3 (b) for IC and 3 (c) for NC.

For WC, $RD_{q_{vert}}$ increases with tree $LAI_{2D}$, $CVF$ and height ratio (Fig. 3(a)). For IC, $RD_{q_{vert}}$ also increases with $CVF$ and height ratio but $RD_{q_{vert}}$ tends to slightly decrease as the LAI increase when the LAI is high and the ratio $h/H$ is small (differences between $LAI = 3$ and $4$ for $h/H = 0.36$ and $0.50$) (Fig. 3(b)). For NC, an small increase of $RD_{q_{vert}}$ with $LAI_{2D}$

and $CVF$ is observed for $h/H = 0.65$. However, for the two other smaller height ratios the tree effect is very low ($-1.1 \leq RD_{q_{vert}} \leq 0.5$) (Fig. 3(c)).

For vertical transfer, a wind perpendicular to the street ($\varphi = 90$ °) is used to focus on the effects on vertical transfers only. The complex air flow occurring in street canyons for a perpendicular wind leads to heterogeneous tracer concentration in the street. In general for the three canyons, the presence of the two tree crowns tends to increase the tracer concentration on the leeward

side of the street and to decrease it on the windward side (Buccolieri et al., 2009; Gromke and Ruck, 2012). Depending on street aspect ratio and therefore on the air flow regime (Oke, 1988; Harman et al., 2004) as well as on the tree characteristics, the tree crown effect on local tracer concentration (leeward versus windward side) is more or less important. For example, for NC, the flow regime is skimming, and in average, the variation of $C_{street}$ due to the presence of trees compensates between the two sides of the street explaining why $RD_{q_{vert}}$ is very low for NC.

Note that unlike $U_{street}$ (Sect. 3.1), the tree effect on $q_{vert}$ decreases when the canyon gets deeper. For example, the tree effect, as quantified by $RD_{q_{vert}}$, ranges between -1.0 and -20.3 % for WC, -1.0 and -18.7 % for IC and 0.5 and -2.7 % for NC. The averaged tree effect is relatively less strong when the canyon is deeper. In other words, regarding vertical transfers the street effect dominates over the tree effect.

This section demonstrated that the tree effect parameterizations should depend on tree characteristics and also on building

characteristics to account for the building-tree interactions. The next section aims to parameterize in MUNICH the tree effect on $U_{street}$ and $q_{vert}$ observed in Code_Saturne simulations.

**Figure 3.** Relative deviation RD of $q_{vert}$ computed from Code_Saturne simulations for different tree $LAI_{2D}$, $CVF$ and height ratio for WC (a), IC (b) and NC (c). The graphic is divided into three columns corresponding to the three height ratios, and higher $CVF$ correspond to darker colors. $|RD_{q_{vert}}|$ values are specified with data labels.

## 4 Parameterization of the aerodynamic effect of tree crowns in MUNICH

### 4.1 Model description

The MUNICH parameterizations of horizontal and vertical transfers detailed in Maison et al. (2022) are modified to take into

account the tree effects. These parameterizations are based on Wang (2012, 2014) equations, which were originally developed for homogeneous vegetated cover. To remain consistent with this hypothesis, the parameterization will depend on the homo-





geneous Leaf Area Index in the street noted $LAI_{street}$. As for the conversion from $LAI_{3D}$ to $LAI_{2D}$, $LAI_{street}$ is estimated from $LAI_{2D}$ conserving the leaf surface. $LAI_{street}$ is calculated by spreading the tree crown cylindrical $LAI_{2D}$ over the whole street width:

$$LAI_{street} \times S_{street} = LAI_{2D} \times S_{2D} \tag{17}$$

$$\implies LAI_{street} = LAI_{2D} \times \frac{S_{2D}}{S_{street}} = \frac{2rLn\,LAI_{2D}}{WL} = \frac{2r\,n\,LAI_{2D}}{W}, \tag{18}$$

where $S_{street}$ is the soil projected area of the street homogeneous tree crown and $S_{2D}$ is the soil projected area of the Code_Saturne 2D cylindrical tree crown (in m$^2$). Note that regardless of the tree crown geometry, $LAI_{street}$ is always equal to the street total leaf surface divided by the street ground area ($WL$).

To account for the tree effect on the mixing length, the characteristic length of the trees ($l_{c_t}$ in m) is added into the equation describing the mixing length $l_m$:

$$\frac{1}{l_m} = \frac{1}{\kappa H} + \frac{1}{l_{c_b}} + \frac{1}{l_{c_t}\,f_{b \times t}}, \tag{19}$$

$$\text{with} \quad l_{c_t} = \frac{E_t H}{C_{D_t}\,\frac{1}{2}LAI_{street}}, \tag{20}$$

where $\frac{1}{2}LAI_{street}$ corresponds to the leaf frontal area density assuming a random leaf orientation distribution, $E_t$ is a proportionality constant taken equal to 0.054 for vegetated cover as suggested by Wang (2014), and $f_{b \times t}$ is a function parameterized based on Code_Saturne simulations in Sect. 4.2 and representing the interaction between buildings, trees and the tree crown height.

The dimensionless factor $s_H$ expressing the effects of the tree canopy on the mixing length $l_m$ is calculated using its definition in Eq. (6) and the expression of $l_m$ in Eq. (19):

$$l_m = \kappa H s_H \quad \text{at roof level} \tag{21}$$

$$\text{with } s_H = \begin{cases} \frac{l_{c_b}}{l_{c_b} + \kappa H} & \text{without tree} \\ \frac{l_{c_b}\,l_{c_t}\,f_{b \times t}}{\kappa H \left(l_{c_b} + l_{c_t}f_{b \times t}\right) + l_{c_b}\,l_{c_t}\,f_{b \times t}} & \text{with trees.} \end{cases} \tag{22}$$

The simulations with and without trees have to be distinguished to avoid any convergence issue since $l_{c_t}$ tends to $+\infty$ when $LAI_{street}$ tends to 0. This $s_H$ factor includes now rough soil, building and tree effects on mixing length and is then used in the calculation of the attenuation coefficient $\alpha$ (Eq. (2)). In the numerator, the expressions for buildings and trees are added:

$$\alpha = \frac{C_B\,a_r + C_{D_t}\,C_u\,\frac{1}{2}LAI_{street}}{\kappa\,s_H}, \tag{23}$$

where $C_{D_t}$ is the tree drag coefficient (dimensionless) taken equal to the one used in Code_Saturne ($C_{D_t} = 0.2$). Wang (2014) presents $C_u$ as a dimensionless coefficient homogeneous on the vertical axis, but that can depend on canopy features. This coefficient has to be determined based on observation data and will be parameterized in Sect. 4.2.





### 4.2 Parameter determination based on Code_Saturne simulations

Two parameters introduced in $l_m$ and $\alpha$ equations (Eq. (19) and (23)) have to be determined based on Code_Saturne: $RD_{q_{vert}}$ and $RD_{U_{street}}$. The function $f_{b \times t}$ depends on $a_r$ to account for the interaction between trees and buildings observed in Fig. 3 and also on $h_{max}$ to account for the effect of the tree height. Note that since tree crowns are assumed to be homogeneous within the canopy in the original Wang formulation, the tree crown height $h_{max}$ is not taken into account (see Eq. (20)), and therefore it needs to be included in the function $f_{b \times t}$. The $f_{b \times t}$ expression is determined by maximizing the fit between Code_Saturne $RD_{q_{vert}}$ and MUNICH $RD_{q_{vert}}$:

$$f_{b \times t} = \frac{a_0 + a_1 \exp(a_2\, a_r)}{(h_{max}/H)^2} \quad \text{with } a_0 = 3.26,\ a_1 = 0.0256 \text{ and } a_2 = 6.70. \tag{24}$$

A comparison of Code_Saturne $RD_{q_{vert}}$ and the $RD_{q_{vert}}$ parameterized with Eqs. (7), (19), (20) and (24) is presented for the three canyons in Fig. 4 (a,c,e). Normalized Mean Absolute Error (NMAE) and Bias (NMB) are calculated to compare Code_Saturne and MUNICH results (see Appendix C for the definition of the statistical indicators). Figures 4 (a,c,e) show a good agreement between Code_Saturne $RD_{q_{vert}}$ and the parameterized $RD_{q_{vert}}$ because the $f_{b \times t}$ function was determined to minimize bias. The NMAE can be high for $RD_{q_{vert}}$ (up to 63 % in NC) because the parameterization does not reproduce the slight decrease of $RD_{q_{vert}}$ when $LAI_{2D}$ increases from $LAI_{2D} = 3$ to $4$ when $h/H \approx 1/3$ and $1/2$ (see section 3.2, Fig. 3 (b) and (c)). But this is not an issue for NC since the $q_{vert}$ values are low.

For $U_{street}$, the parameter $C_u$ also needs to be determined. A constant value of $C_u = 6.7$ is sufficient to get a good fit between Code_Saturne $RD_{U_{street}}$ and MUNICH $RD_{U_{street}}$. No dependency of $C_u$ on building or tree features is needed, as shown in Fig. 4 (b,d,f), which compares Code_Saturne $RD_{U_{street}}$ and the parameterized $RD_{U_{street}}$. Note that the parameterizations of $l_{c_t}$ and $f_{b \times t}$ impact not only the vertical transfers but also the horizontal wind speed, because they are involved in the calculation of the $s_H$ factor and hence of the $\alpha$ coefficient. The good comparisons between Code_Saturne $RD_{U_{street}}$ and MUNICH $RD_{U_{street}}$ also shows that the parameterized $s_H$ reproduces well the horizontal wind speed, and therefore the interactions between trees and building effects and the influence of $h_{max}$.



**Figure 4.** Comparison of (a,c,e) $RD_{q_{vert}}$ and (b,d,f) $RD_{U_{street}}$ computed from Code_Saturne simulations and parameterized in MUNICH for different tree $LAI_{2D}$, $CVF$ and height ratio and for (a,b) WC, (c,d) IC and (e,f) NC.





## 4.3 Comparison of $q_{vert}$, $U_{street}$ and wind profiles

Figure 5 (a) and (b) presents respectively a comparison of Code_Saturne and parameterized $q_{vert}$ and $U_{street}$. The statistical indicators are presented in Table 2. As MUNICH was parameterized to reproduce well the tree effect observed in Code_Saturne, and as the two models agree well in a treeless canyon (Maison et al., 2022), the vertical transfer coefficient and wind speed with trees are close between the two models as expected (Fig. 5 and Table 2). The parameterized vertical transfer coefficients with trees agree well with Code_Saturne ones with normalized mean absolute errors ranging from 2.2 to 4.1 % and bias from -4.1 to 2.8 %. The parameterized average wind speed with trees agree well with Code_Saturne ones with normalized mean absolute errors ranging from 3.4 to 6.8 % and a normalized mean bias from -2.8 to -1.1 %.

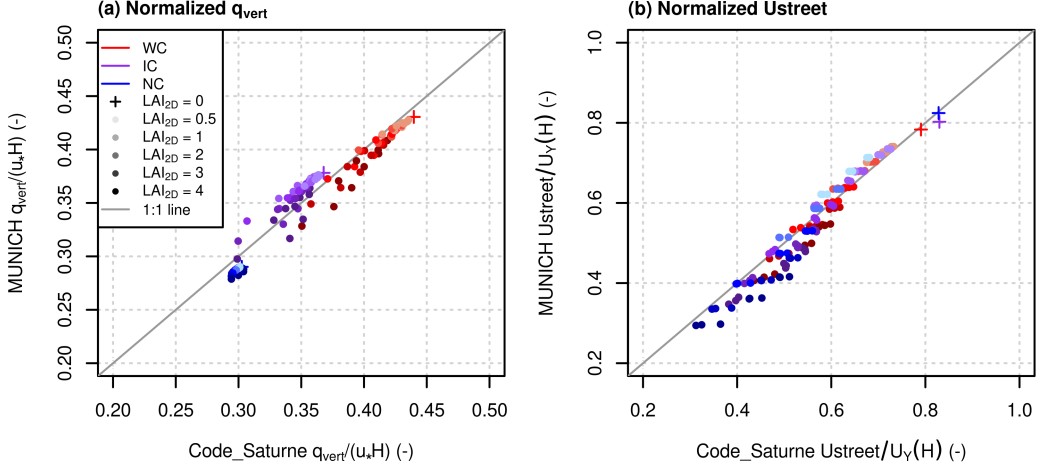

**Figure 5.** Normalized $U_{street}$ and $q_{vert}$ computed from Code_Saturne simulations for different tree $LAI_{2D}$, $CVF$ and height ratios.

**Table 2.** Statistical indicators (Normalized Mean Absolute Error (NMAE) and Bias (NMB) in %) for the comparison of Code_Saturne and MUNICH $q_{vert}$ and $U_{street}$.

| Canyon | $q_{vert}$ | | $U_{street}$ | |
|---|---|---|---|---|
| | NMAE | NMB | NMAE | NMB |
| WC | 2.2 | -2.1 | 3.4 | -1.1 |
| IC | 3.5 | 2.8 | 4.4 | -1.7 |
| NC | 4.1 | -4.1 | 6.8 | -2.8 |

For each street canyon, Fig. 6 compares the Code_Saturne and parameterized vertical wind profiles for fixed $CVF$ and $h/H$ ratio but for five different $LAI_{2D}$. It shows that an increase of the $LAI_{2D}$ induces a decrease of the wind velocity. Concerning the vertical profile shape, trees induce a lower wind velocity on the entire street and not only in the tree crown. They even





slightly impact the velocity just above the street. The maximum of attenuation of the wind is located in the middle of the tree crown and the wind velocity is re-accelerated under the tree crown. In the middle of the tree crown, the parameterized wind speed is close to the one of Code_Saturne. In the lower part and under the tree crown, the parameterized wind speed is under-estimated. In fact, the re-acceleration under the tree crown is complex to consider in parameterized models. Besides, in real streets the trees height is not homogeneous so this re-acceleration under the tree crown might be unrealistic in Code_Saturne simulations. In the parameterized wind profile with trees, the reduction of the wind speed just above the street due to the presence of trees was neglected. So the parameterized wind speed above and in the upper part of the tree crown is overesti-mated compared to Code_Saturne. Note that Kent et al. (2017) proposed a method to include vegetation effect in aerodynamic roughness parameters and so to account for vegetation in the above urban canopy wind profile of Macdonald et al. (1998). This method considers vegetation at the city scale and is not applicable here, since we work at the street scale. For now, as MUNICH assumes an homogeneous street canyon, the profile is averaged to compute $U_{street}$ and the shape of the profile is not used.

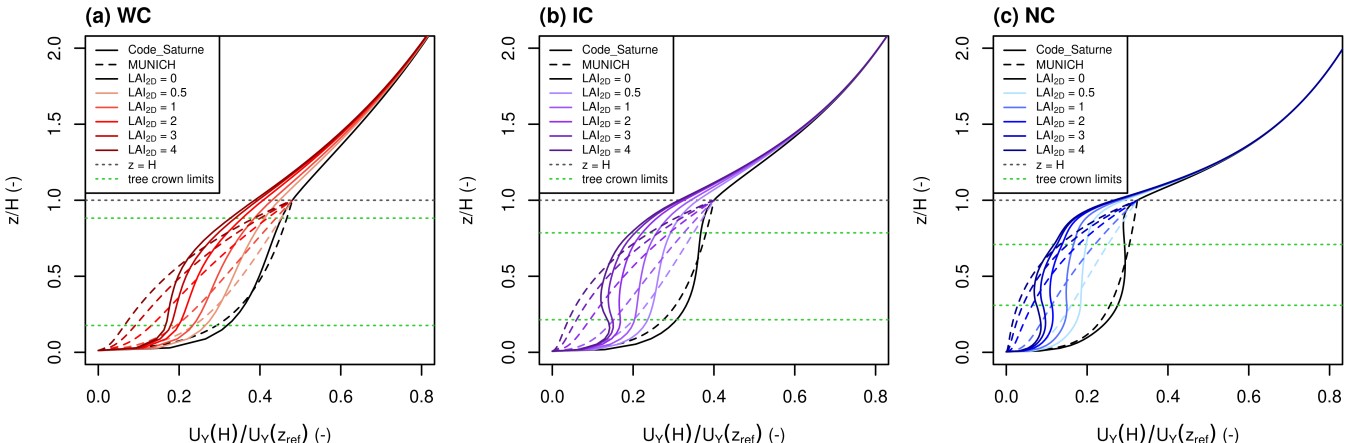

**Figure 6.** Comparison of Code_Saturne (solid lines) and parameterized (dotted lines) vertical wind profiles. The profiles in a treeless canyon are in black and darker colors correspond to increasing $LAI_{2D}$. For the three canyons, $h/H \approx 0.5$ and $CVF \approx 25$ %.

## 5 Conclusions

Although the discretizations and the physical modeling used in MUNICH and Code_Saturne are fundamentally different, the setup of the CFD simulations was adapted to compare the two models through the average horizontal wind speed along the street and the calculation of the vertical transfer coefficient at the roof level by averaging a passive tracer concentration in the street and the background (above the streets). To build a parameterization suited to the street network MUNICH, the CFD sim-ulations were simplified with several hypothesis such as an infinite street canyon with 2D CFD simulations and homogeneous emissions over the street. Furthermore, quantities of interest were averaged over the street in the CFD simulations.

The aerodynamic effect of trees in street canyons is quantified with Code_Saturne simulations. Simulations were performed





with 2 rows of trees and a large range of tree Leaf Area Index, crown radius and heights. In the range of the simulations performed, the average horizontal velocity in the street is reduced by -7.3 to -62.3 % and the vertical transfer coefficient by +0.5 to -20.3 %. This highlights the necessity of adding the tree aerodynamic effects into street models, such as MUNICH.

340 Maison et al. (2022) proposed a parameterization of horizontal and vertical transfers in a treeless canyon and the present study adds the tree effect to the building effect in this parameterization. The differences of wind speed and vertical transfer coefficient in a treeless canyon are very low between the two models (Maison et al., 2022). However, to overcome any model-specific difference and to build a parameterization for MUNICH of the tree effect on the aerodynamic parameters, relative deviation ratios between cases without and with trees are defined.

345 First, the Maison et al. (2022) vertical transfer coefficient parameterization is modified by adding a term representing the tree effect and the interaction between the trees and the buildings in the mixing-length expression. This term is function of two dimensionless parameters to characterize the trees: street Leaf Area Index ($LAI_{street}$) and tree to street height ratio ($h_{max}/H$) as well as building height ($H$) and street width ($W$). A parameterization of the tree mixing length was defined from Code_Saturne simulations to obtain a tree effect as close as possible in the two models. The parameterized vertical transfer coefficients with 350 trees agree well with those of Code_Saturne with normalized mean absolute error ranging from 2.2 to 4.1 % and bias from -4.1 to 2.8 %.

Second, this new mixing length expression is also used to compute the vertical wind profile and average wind speed along the street. Only one constant is fixed in the wind attenuation coefficient to maximize the fit between Code_Saturne and MUNICH tree effect on average wind speed. The comparison of the average wind speed gives a normalized mean absolute error ranging 355 from 3.4 to 6.8 % and a normalized mean bias from -2.8 to -1.1 %.

MUNICH now includes a relatively simple parameterization of tree effect on both horizontal and vertical aerodynamic processes based on commonly used tree parameters, that can be easily computed from urban databases and can reproduce the main effects obtained in much more detailed and costly CFD simulations. This parameterization can also be used in urban climate models to compute water and heat transfers in tree-lined streets. The perspectives of this study are to quantify the effect of street 360 trees on air quality from the street level to the scale of the city of Paris and to compare biogenic and anthropic contributions to air pollution over an entire city with the fast running code MUNICH.

*Code availability.* The last version of MUNICH source code is available online at https://doi.org/10.5281/zenodo.4168984 and https://github.com/cerea-lab/munich (last access on 5 January 2022).

For the three 2D canyons considered in the present study, the mesh, the source code and the XML setup file allowing reproduction of 365 the CFD results using Code_Saturne version 6.0 are available online at https://gitlab.enpc.fr/alice.maison/tree_parametrization and at http://dx.doi.org/10.17632/fzfrjsz3mv.2 under the GNU GPL2.0 licence (last access on 3 December 2021).





# Appendix A: Lists of abbreviations, variables parameters and tree dimensions

**Table A1.** List of abbreviations

| Acronym | Definition |
| --- | --- |
| WC | Wide canyon |
| IC | Intermediate canyon |
| NC | Narrow canyon |
| CTM | Chemistry-transport model |
| CFD | Computational fluid dynamics |
| RD | Relative deviation |

**Table A2.** List of parameters

| Symbol | Definition | Value | Unit |
| --- | --- | --- | --- |
| $\kappa$ | Von Kàrmàn constant | 0.42 | - |
| PBLH | Planetary Boundary Layer Height | 1000 | m |
| $z_{0_s}$ | Code_Saturne inside street walls roughness length | 0.10 | m |
| $u_*$ | Friction velocity | 0.727 | m.s$^{-1}$ |
| $E$ | Parameter in modified Wang (2014) parameterization | 0.5 | - |
| $n$ | Number of tree rows | 2 | - |
| $C_{D_t}$ | Tree crown drag coefficient (Katul and Albertson, 1998) | 0.2 | - |



**Table A3.** List of variables

| Group of variables | Symbol | Definition | Unit |
|---|---|---|---|
| Street characteristics | $H$ | Buildings height | m |
| | $W$ | Street width | m |
| | $L$ | Street length | m |
| | $a_r$ | Street aspect ratio | - |
| | $l_{c_b}$ | Characteristic length in the street | m |
| Tree characteristics | $LAD$ | Tree Leaf Area Density | $m^2_{leaf}.m^{-3}_{tree\ crown}$ |
| | $LAI$ | Tree Leaf Area Index | $m^2_{leaf}.m^{-2}_{soil}$ |
| | $LAI_{3D}$ | 3D Leaf Area Index of the spherical tree crown | $m^2_{leaf}.m^{-2}_{soil}$ |
| | $LAI_{2D}$ | 2D equivalent Leaf Area Index of the cylindrical tree crown | $m^2_{leaf}.m^{-2}_{soil}$ |
| | $LAI_{street}$ | Leaf Area Index of the homogeneous street tree crown | $m^2_{leaf}.m^{-2}_{soil}$ |
| | $S_{3D}$ | Soil projected area of the 3D spherical tree crown | $m^2$ |
| | $S_{2D}$ | Soil projected area of the 2D cylindrical tree crown | $m^2$ |
| | $S_{street}$ | Soil projected area of the street homogeneous tree crown | $m^2$ |
| | $r$ | Tree radius | m |
| | $CVF$ | Crown Volume Fraction | - or % |
| | $h$ | Middle crown height | m |
| | $h_{max}$ | Maximum tree crown height | m |
| | $h_{min}$ | Minimum tree crown height | m |
| | $\delta$ | Spacing between two trees within a row | m |
| | $l_{c_t}$ | Tree characteristic length | m |
| Horizontal wind speed | $U_{street}$ | Average street horizontal wind speed | $m.s^{-1}$ |
| | $U$ | Norm of the horizontal wind speed | $m.s^{-1}$ |
| | $U_X$ | Horizontal wind speed in the $X$ direction | $m.s^{-1}$ |
| | $U_Y$ | Horizontal wind speed in the $Y$ direction | $m.s^{-1}$ |
| | $U_H$ | Average horizontal wind speed at the roof level | $m.s^{-1}$ |
| | $\varphi$ | Angle of the wind direction | rad or ° |
| | $\alpha$ | Wind attenuation coefficient | - |
| | $s_H$ | Characteristic length factor | - |
| | $C_u$ | Empiric coefficient in $\alpha$ equation | - |
| | $C_B$ | Function of $a_r$ and $\varphi$ | - |
| Vertical transfer | $Q_{vert}$ | Vertical flux of pollutant | $\mu g.s^{-1}$ |
| | $q_{vert}$ | Vertical transfer coefficient | $m^2.s^{-1}$ |
| | $\sigma_W$ | Standard deviation of the vertical wind speed at $z = H$ (Soulhac et al., 2011) | $m.s^{-1}$ |
| | $l_m$ | Mixing length in the street | m |
| | $e$ | Passive tracer emission rate for the street | $\mu g.s^{-1}$ |
| | $C_{street}$ | Street concentration | $\mu g.m^{-3}$ |
| | $C_{bg}$ | Background concentration | $\mu g.m^{-3}$ |





**Table A4.** List of tree dimensions simulated with Code_Saturne ($r$: radius, $h$: middle crown height, $h_{min}$: crown bottom height $h_{max}$: crown top height) and calculated tree parameters ($CVF$: crown volume fraction and $h/H$: height ratio).

| Canyon | $r$ (m) | $h$ (m) | $h_{min}$ (m) | $h_{max}$ (m) | $CVF$ (%) | $h/H$ (-) |
|--------|--------|--------|--------|--------|--------|--------|
|     | 1.5 | 3.5 | 2.0 | 5.0 | 6.0 | 0.41 |
|     | 1.5 | 4.5 | 3.0 | 6.0 | 6.0 | 0.53 |
|     | 1.5 | 5.5 | 4.0 | 7.0 | 6.0 | 0.65 |
|     | 2.0 | 3.5 | 1.5 | 5.5 | 10.8 | 0.41 |
| WC  | 2.0 | 4.5 | 2.5 | 6.5 | 10.8 | 0.53 |
|     | 2.0 | 5.5 | 3.5 | 7.5 | 10.8 | 0.65 |
|     | 3.0 | 3.5 | 0.5 | 6.5 | 24.2 | 0.41 |
|     | 3.0 | 4.5 | 1.5 | 7.5 | 24.2 | 0.53 |
|     | 3.0 | 5.5 | 2.5 | 8.5 | 24.2 | 0.65 |
|     | 2.0 | 5.0 | 3.0 | 7.0 | 6.5 | 0.36 |
|     | 2.0 | 7.0 | 5.0 | 9.0 | 6.5 | 0.50 |
|     | 2.0 | 9.0 | 7.0 | 11.0 | 6.5 | 0.64 |
|     | 2.5 | 5.0 | 2.5 | 7.5 | 10.2 | 0.36 |
| IC  | 2.5 | 7.0 | 4.5 | 9.5 | 10.2 | 0.50 |
|     | 2.5 | 9.0 | 6.5 | 11.5 | 10.2 | 0.64 |
|     | 4.0 | 5.0 | 1.0 | 9.0 | 26.1 | 0.36 |
|     | 4.0 | 7.0 | 3.0 | 10.0 | 26.1 | 0.50 |
|     | 4.0 | 9.0 | 5.0 | 13.0 | 26.1 | 0.64 |
|     | 2.5 | 9.0 | 6.5 | 11.5 | 5.2 | 0.33 |
|     | 2.5 | 14.0 | 11.5 | 16.5 | 5.2 | 0.51 |
|     | 2.5 | 18.0 | 15.5 | 20.5 | 5.2 | 0.65 |
|     | 3.5 | 9.0 | 5.5 | 12.5 | 10.2 | 0.33 |
| NC  | 3.5 | 14.0 | 10.5 | 17.5 | 10.2 | 0.51 |
|     | 3.5 | 18.0 | 14.5 | 21.5 | 10.2 | 0.65 |
|     | 5.5 | 9.0 | 3.5 | 14.5 | 25.1 | 0.33 |
|     | 5.5 | 14.0 | 8.5 | 19.5 | 25.1 | 0.51 |
|     | 5.5 | 18.0 | 12.5 | 23.5 | 25.1 | 0.65 |

Each line of the Table A4 corresponds to the parameters of a simulation. For each case, simulations with different $LAI_{2D}$ were performed: 0.5, 1, 2, 3 and 4.



## Appendix B: Comparison of 2D and 3D Code_Saturne simulations with trees

To save computing time, Code_Saturne simulations were performed in a periodic 2D canyon of length $L = 1$ m, and the street was therefore considered infinite due to this periodicity. In this case, the tree crown is represented as an infinite cylinder of radius $r$. To take into account the fact that in reality most of the tree crowns are spherical and are spaced from each other, an equivalent cylindrical $LAI_{2D}$ is calculated in the 2D simulations. For the tree effect to be similar, the surface of leaves is kept constant between the 2D and 3D simulations:

$$LAI_{3D} \times S_{3D} = LAI_{2D} \times S_{2D} \tag{B1}$$

$$LAI_{2D} = LAI_{3D} \times \frac{S_{3D}}{S_{2D}} = LAI_{3D} \times \frac{\pi r^2}{2r\,(2r + \delta)}, \tag{B2}$$

where $\delta$ is the spacing between two tree crowns within the row and $2r + \delta$ is the length of the street section for one spherical tree (m). $S_{3D}$ and $S_{2D}$ are respectively the soil projected area of the $3D$ and $2D$ tree crowns (m$^2$). In addition to the leaf surface and tree height, the tree radius is also conserved between 2D and 3D simulations, however the value of the Crown Volume Fraction ($CVF$) varies:

$$CVF_{2D} = \frac{V_{2D}}{V_{street}} = \frac{n\pi r^2 L}{HWL} \quad \text{and} \quad CVF_{3D} = \frac{V_{3D}}{V_{street}} = \frac{n\frac{4}{3}\pi r^3}{HW\,(2r + \delta)} \tag{B3}$$

$V_{street}$ is the street volume (m$^3$), $V_{3D}$ and $V_{2D}$ are respectively the $3D$ and $2D$ tree crown volumes (m$^3$). The relation between 2D and 3D $CVF$ is therefore:

$$CVF_{2D} = \frac{3}{4} \times \frac{2r + \delta}{r} \times CVF_{3D} \tag{B4}$$

2D and 3D simulations with different $LAI$ are performed to check the validity of the 2D infinite street canyon hypothesis. 3D simulations are realized with a periodic street of 10 m-length and cell meshes of $0.5 \times 0.5 \times 0.5$ m in the $X$, $Y$ and $Z$ directions. In both 2D and 3D simulations, $C_{street}$ and $C_{bg}$ are computed with a normal wind to the street ($\varphi = 90$ °) and $U_{street}$ with $\varphi = 45$ °. The results are presented in Fig. B1 (a-d) for WC, (e-h) for IC and (i-l) for NC.

For each canyon, the cases without tree ($LAI = 0$) are equivalent: $C_{street}$, $C_{bg}$, $q_{vert}$ and $U_{street}$ are similar between the 2D and 3D cases with a maximum relative difference of 0.1 % for $U_{street}$. For the cases with trees, concerning $C_{street}$, $C_{bg}$ and $q_{vert}$, the differences between the 2D and 3D cases are relatively low over the range of simulations performed with a maximum of 3.3 % for $C_{street}$ and $C_{bg}$ in WC. For $U_{street}$, slightly larger differences are observed (up to $-5.4$ % in WC for $LAI = 4(2D)/12.7(3D)$). These differences can be explained by the distribution of the turbulence around the crown. For each variable and canyon, the relative deviation between the 2D and 3D cases increases with the tree $LAI$. Given the low differences observed between the 2D and 3D simulations, the hypothesis of a 2D canyon with a cylindrical tree crown associated with an equivalent $LAI_{2D}$ to represent a spherical tree crown spaced of $\delta$ m in a street of length $L$ is reasonable.



**Figure B1.** Comparison of $C_{street}$ (a,e,i), $C_{bg}$ (b,f,j), $q_{vert}$ (c,g,k) and $U_{street}$ (d,h,l) for 2D and 3D simulations in the three canyons without and with trees of $LAI_{2D} = 1$ and 4. $CVF_{2D} \approx 10\,\%$ and $h/H \approx 0.5$ are fixed. The percentages written in the middle of the histogram bars correspond to the RD between the 2D and 3D cases.



## Appendix C: Definition of the statistical indicators

Code_Saturne and MUNICH simulations are noted $cs_i$ and $m_i$, respectively. In this section, $n$ is the total number of simulations which is equal to 45 per street canyon.

- Normalized Mean Absolute Error (%):

$$NMAE = 100 \times \frac{\sum\limits_{i=1}^{n} |m_i - cs_i|}{\left| \sum\limits_{i=1}^{n} cs_i \right|} \tag{C1}$$

- Normalized Mean Bias (%):

$$NMB = 100 \times \frac{\sum\limits_{i=1}^{n} (m_i - cs_i)}{\sum\limits_{i=1}^{n} cs_i} \tag{C2}$$

*Author contributions.* KS, CF, BC and AM were responsible for conceptualization. AM developed the MUNICH model code and performed the Code_Saturne simulations. AM, KS, CF and BC performed the formal analysis. AM conducted the visualization. AM, KS and CF were responsible for writing the original draft and BC and AT reviewed it. CF, BC and YW provided support in Code_Saturne computing resources. KS and AT were responsible for funding acquisition.

*Competing interests.* The authors declare that they have no conflict of interest.

*Acknowledgements.* This work was partially funded by the sTREEt ANR project (ANR-19-CE22-0012). The authors thank Youngseob KIM and Lya LUGON for their support in the development of the MUNICH model and Martin FERRAND for his support in the understanding of the Code_Saturne model.



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
