# Peer review of "Parameterizing the aerodynamic effect of trees in street canyons for the street-network model MUNICH using the CFD model Code\_Saturne"

_Atmospheric Chemistry and Physics, 2022_

## Author Response (AR1)

Atmos. Chem. Phys. Discuss., author comment AC1
https://doi.org/10.5194/acp-2022-287-AC1, 2022

[Figure]

**Reply on RC1**

Alice Maison et al.

Author comment on "Parameterizing the aerodynamic effect of trees in street canyons for the street-network model MUNICH using the CFD model Code_Saturne" by Alice Maison et al., Atmos. Chem. Phys. Discuss., https://doi.org/10.5194/acp-2022-287-AC1, 2022

The paper proposed a parameterization to include the aerodynamic effect of tree crowns in the MUNICH street-network model. For that, simulations using Code_Saturne CFD code are performed varying the Leaf Area Index, the crown volume fraction, and the ratio between tree and street height for three types of street canyons. The effects of these parameters over the horizontal velocity (Ustreet) and vertical transfer coefficient (qvert) in treeless and tree conditions are studied. Then, these effects are included in MUNICH horizontal and vertical transfer parameterization from Maison et al. (2022) which is based on Wang (2012, 2014). The qvert and Ustreet calculated with the new MUNICH parameterization including trees agree with Code_Saturne simulations. The parameterized wind profile is closer to Code_Saturne for winds near the middle of the tree crown. The paper is well structured and written. The assumptions are well explained, with a clear methodology, interesting results which are well supported by the plots, and conclusions in concordance with the journal scope. This work represents an improvement on MUNICH formulations with the potential of improving air quality simulation inside urban canyons. I think this paper is also a good example of the process of building new parameterization in street-network models.

Minor revisions are detailed below:

Specific comments:

- Line 106. A definition of phi, the angle formed by the wind direction above the roof and the direction of the street, is missing.

**Authors' response:** The definition of phi is added line 106.

- Further details on Code_Saturne configuration are required. Specifically, What are the

criteria to determine the location of the trees (7m from the wall)?

**Authors' response:** The following sentence is added (line 142): "The tree crown centers are located in a position on the X axis so that the largest crowns (CVF ≈ 25%) do not reach the street walls for the three street canyons studied."

- What are the criteria to determine the maximum height of the Domain?

**Authors' response:** The maximum height of the domain is equal to 3 times the building height ("Maximum height of the domain **(3H)**" is added in table 1). This factor of three on the mean building height is usually considered the height of the urban canopy (or the "Roughness Sub Layer") where the flow is highly spatially variable (Roth 2000).

- It is not clear if the reason for using a two-dimensional set-up is only to save computational time or is it a required adaptation to compare Code_Saturne simulation with MUNICH (in line 89).

**Authors' response:** MUNICH corresponds to a simplified representation of the geometry of street canyons and considers a network of segments of homogeneous street in the street direction (Y axis). It is thus necessary to perform CFD studies using simplified geometries to stay within this generic approach. So the main reason for using a 2D setup is to compare Code_Saturne simulations with MUNICH on those street segments. Some 3D CFD computations have also been performed for comparison purposes (not shown).

In Appendix 3 (line 371) "To save computing time" is removed.

- Fig 1. The location of the background and location of the inlet can be added in Fig. 1. Like in Fig 1. in Maison et al. (2022)

**Authors' response:** Background and location of the inlet and outlet borders are added on Fig. 1.

- Line 232-236. More details should be given on the reasons for the reduction of the effect on qvert (RD_qvert) in Intermediate and Narrow Canyons with high LAI values and small h/H ratio ( Fig 3(b) and Fig 3(c)).

**Authors' response:** These reductions of RD_qvert in IC and NC with high LAI values and small h/H ratios are due to differences in detailed micro-scale flow and turbulence changes.

In the intermediate canyon, when the LAI increases, the wind speed decreases at the tree crown level but increases between the tree crowns. When the h/H ratio is low (h/H = 0.36 and 0.5), this reacceleration is located at the bottom leewards side of the street where the tracer concentrations are the highest. This leads to an increase of the tracer dispersion and a decrease in the street average concentration. Therefore, an increase of LAI induces a lower RD_qvert.

In the narrow canyon, this variation of qvert due to tree effects can also be related to the

turbulent viscosity. The turbulent viscosity is decreased at the crown level and under the tree crown but is increased above the tree crown. When the h/H ratio is low, this area above the tree crown where the turbulent viscosity is increased is larger than for higher h/H ratios leading to more vertical transfer between the street and the background. This induces slightly lower street average concentration and a reduced effect of the tree crown on qvert (RD_qvert).

Overall, these variations due to micro-scale effects are relatively low (0 to 3%) and have a negligible impact on the global effect of trees in street canyons.

To discuss this issue, the following paragraph is added in the article: "This reduction of the tree effect on qvert when LAI or CVF increases for small h/H ratios in IC and NC can be explained by micro-scale effects (modified air flow path and turbulent viscosity) and is left out of the scope of the present study due to the corresponding low amplitude. Besides, investigation of such micro-scale effects would probably require more advanced turbulence models (switching from a first-order model (k-□□) to a second-order one (Rij-SSG (Speziale et al., 1991)), or a 3D Large Eddy Simulation (LES))."

- Fig 4. What is the meaning of ER in Fig 4 subtitles? It can be added to the figure label.

**Authors' response:** It is RD (relative deviation) and not ER, the figure subtitles are corrected.

Technical corrections:

- For consistency, Eq (15) and Eq (16) are missing the " x 100" as it is expressed in % in the plots and analysis and also in NMAE and NMB equations.

**Authors' response:** Equations (15) and (16) are corrected.

- Line 280: Eq. (2) should be Eq. (4)

**Authors' response:** The reference is corrected.

- Line 286: The two introduced parameters to be determined based on Code_Saturne are f_bxt and Cu.

**Authors' response:** The parameters f_bxt and Cu are added Line 286.

- Line 333: It should be "street-network model MUNICH".

**Authors' response:** Line 333 is corrected.

References:

Roth, M. (2000). Review of atmospheric turbulence over cities. Quarterly Journal of the Royal Meteorological Society, 126(564), 941-990. https://doi.org/10.1002/qj.49712656409

[Figure]

Atmos. Chem. Phys. Discuss., author comment AC2
https://doi.org/10.5194/acp-2022-287-AC2, 2022

[Figure]

**Reply on RC2**

Alice Maison et al.

Author comment on "Parameterizing the aerodynamic effect of trees in street canyons for the street-network model MUNICH using the CFD model Code_Saturne" by Alice Maison et al., Atmos. Chem. Phys. Discuss., https://doi.org/10.5194/acp-2022-287-AC2, 2022

Maison et al have proposed a simple parametrization to account for the effect of trees in street canyons for the MUNICH model. The authors made a comparison between the MUNICH parametrization and the Code Saturn as shown in figure 4 and table 2. Hence, the proposed parametrizations seem reasonable. Evaluating the effect of the ecosystemic services that urban trees provide is essential for climate change, air pollution, and well-being. Actually, in the introduction Maison et al it is shown contradictory results about the positive or negative effects of urban trees. The abstract shows that the tree crown slowed down the flow and reduced the average horizontal velocity up to 68%.

Main issues:

- Then, my main question, is why the authors are not providing a comparison of air pollution of MUNICH with and without this new parametrization? Please, include this comparison for a real case. This type of comparison will be beneficial for policymaking.

**Authors' response:** The purpose of this article is to describe in detail the tree aerodynamic parameterization. In the future, we plan to compare the urban tree impacts on air quality more complex urban setting by taking into account the tree crown aerodynamic effect (using the parameterizations developed in the present paper) but also the radiative effect, the dry deposition on tree leaves and the biogenic organic volatile compounds emitted by the trees. However, the combined effects of trees are complex to model, and this can not be done in a paragraph of this paper. For example, the impact of dry deposition differs completely depending on the pollutant considered (it is highly dependent on the volatility and solubility of the pollutant). In future work, the combined effects will be taken into account and simulations will be compared to measurements results collected in a real case in the city of Paris. We hope this study will help policymakers in their choice of urban tree species and management.

To illustrate the impact of the aerodynamic effect of trees on concentrations, simulations are set up by emitting carbon monoxide (considered as an inert gaseous pollutant) in the street. These simulations are presented and discussed in Appendix D.

Minor issues:

- In general, the article is well written. However, it is not clear if MUNICH includes deposition. If not, any plans?

**Authors' response:** MUNICH includes dry deposition of gaseous pollutants and aerosols on walls and street ground (Kim et al., 2022; Lugon et al., 2021). The dry deposition on tree leaves has been implemented but it is not published yet. Existing parameterizations based on a resistive approach are used for gas (Hicks et al., 1987; Walmsley and Wesely, 1996; Wesely, 1987; Zhang et al., 2002, 2003) and for aerosols (Giardina and Buffa, 2018; Zhang, 2001).

We mentioned at the end of the conclusion that the dry deposition of gaseous pollutants and aerosols on tree leaves will be considered in MUNICH in future studies (see the response to the next question).

- I believe the authors need to explore more the benefits of urban trees and attempt to close the discussion started in the introduction.

**Authors' response:** This paper focuses on building a parameterisation of the aerodynamic effect of trees, so that the model can be used over cities. Other effects, such as dry deposition, thermo-radiative effects and emissions of volatile organic compounds will be taken into account in a future study, where the parameterisation built here will be applied. The conclusion is modified to give more details about this perspective (Line 359):

"The perspectives of this study are to quantify the effect of street trees on air quality from the street level to the scale of the city of Paris. Dry deposition of gaseous pollutants and aerosols on tree leaves as well as emission of biogenic organic volatile compounds related to tree water stress will be considered in MUNICH. The contribution of biogenic and anthropic precursors to the formation of organic aerosols over an entire city will be compared."

- Please, considering discuss these articles:

https://link.springer.com/chapter/10.1007/978-3-319-97013-4_8

https://www.sciencedirect.com/science/article/pii/S1618866706000173?casa_token=HwWC_pLGhWcAAAAA:pGyo5L5EjxJ4eOz9TOifebygB70OtepGHXbEVVLtMo5g7dM-ug4yANgARo9zA_639IKNWJanWJc#!

**Authors' response:** The issue of pollutant deposition on leaves is discussed Lines 47-50 and the reference Nowak et al., (2006) which was just before is moved in this paragraph.

We added Line 50 a sentence to discuss BVOC emissions: "Trees may also affect atmospheric chemistry by emitting biogenic volatile organic compound (BVOC), which may lead to the formation of ozone and secondary organic aerosols (Calfapietra et al., 2013, Prendez et al., 2019, Gu et al., 2021)".

References:

Giardina, M., & Buffa, P. (2018). A new approach for modeling dry deposition velocity of particles. Atmospheric Environment, 180, 11–22. https://doi.org/10.1016/j.atmosenv.2018.02.038

Hicks, B. B., Baldocchi, D. D., Meyers, T. P., Hosker, R. P., & Matt, D. R. (1987). A preliminary multiple resistance routine for deriving dry deposition velocities from measured quantities. Water, Air, and Soil Pollution, 36(3–4), 311–330. https://doi.org/10.1007/BF00229675

Kim, Y., Lugon, L., Maison, A., Sarica, T., Roustan, Y., Valari, M., Zhang, Y., André, M., & Sartelet, K. (2022). MUNICH v2.0: A street-network model coupled with SSH-aerosol (v1.2) for multi-pollutant modelling [Preprint]. Atmospheric sciences. https://doi.org/10.5194/gmd-2022-26

Lugon, L., Vigneron, J., Debert, C., Chrétien, O., & Sartelet, K. (2021). Black carbon modeling in urban areas: Investigating the influence of resuspension and non-exhaust emissions in streets using the Street-in-Grid model for inert particles (SinG-inert). Geoscientific Model Development, 14(11), 7001–7019. https://doi.org/10.5194/gmd-14-7001-2021

Nowak, D. J., Crane, D. E., & Stevens, J. C. (2006). Air pollution removal by urban trees and shrubs in the United States. Urban Forestry & Urban Greening, 4(3–4), 115–123. https://doi.org/10.1016/j.ufug.2006.01.007

Préndez, M., Araya, M., Criollo, C., Egas, C., Farías, I., Fuentealba, R., & González, E. (2019). Urban Trees and Their Relationship with Air Pollution by Particulate Matter and Ozone in Santiago, Chile. In C. Henríquez & H. Romero (Eds.), Urban Climates in Latin America, 167–206. Springer International Publishing. https://doi.org/10.1007/978-3-319-97013-4_8

Walmsley, J. L., & Wesely, M. L. (1996). Modification of coded parametrizations of surface resistances to gaseous dry deposition. Atmospheric Environment, 30(7), 1181–1188. https://doi.org/10.1016/1352-2310(95)00403-3

Wesely, M. L. (1987). Parameterization of surface resistances to gaseous dry deposition in regional-scale numerical models. Atmospheric Environment, 23(6), 1293–1304.

Zhang, L. (2001). A size-segregated particle dry deposition scheme for an atmospheric aerosol module. Atmospheric Environment, 35(3), 549–560. https://doi.org/10.1016/S1352-2310(00)00326-5

Zhang, L., Brook, J. R., & Vet, R. (2003). A revised parameterization for gaseous dry deposition in air-quality models. Atmos. Chem. Phys., 16.

Zhang, L., Moran, M. D., Makar, P. A., Brook, J. R., & Gong, S. (2002). Modelling gaseous dry deposition in AURAMS: A unified regional air-quality modelling system. Atmospheric Environment, 36(3), 537–560. https://doi.org/10.1016/S1352-2310(01)00447-2